# The efficacy and safety of sodium nitroprusside in the treatment of schizophrenia: Protocol for an updated systematic review and meta-analysis

Xinxing Fei[1‡], Shiqi Wang[2,3‡], Jiyang Li[4], Jianxiong Wang[4], Yaqian Gao[5]*, Yue Hu[4]*

**1** Department of Psychiatry, Chengdu Eighth People's Hospital (Geriatric Hospital of Chengdu Medical College), Chengdu, Sichuan, People's Republic of China, **2** Rehabilitation Medicine Center and Institute of Rehabilitation Medicine, West China Hospital, Sichuan University, Chengdu, People's Republic of China, **3** Key Laboratory of Rehabilitation Medicine in Sichuan Province, Chengdu, Sichuan, People's Republic of China, **4** Department of Rehabilitation Medicine, The Affiliated Hospital of Southwest Medical University, Luzhou, Sichuan, People's Republic of China, **5** Department of Rehabilitation Medicine, The First Affiliated Hospital of Chengdu Medical College, Chengdu, Sichuan, People's Republic of China

‡ XF and SW contributed to the work equally and should be regarded as co-first authors.
* 673509366@qq.com (YG); huyue@swmu.edu.cn (YH)

**Data Availability Statement:** No datasets were generated or analysed during the current study. All relevant data from this study will be made available upon study completion.

## Abstract

### Background

Schizophrenia is a chronic persistent disease with high recurrence rate and high disability rate in the field of psychiatry. Sodium nitroprusside is a nitric oxide (NO) donor and considered a promising new compound for the treatment of schizophrenia. New high-quality clinical trials of sodium nitroprusside in the treatment of schizophrenia have been published in recent years. It is necessary to re-conduct the meta-analysis after the inclusion of these new clinical trials. Our study will conduct a systematic review and meta-analysis of the relevant literature in this field, so as to lay an evidence-based medicine foundation for the efficacy of sodium nitroprusside in the treatment of schizophrenia.

### Methods and analysis

Randomized controlled trials (RCTs) of sodium nitroprusside in the treatment of schizophrenia were searched through English databases (PubMed, Web of Science, Embase, and Cochrane Library) and Chinese databases (China Biology Medicine disc, VIP, WanFang Data, and CNKI). The extracted data will be inputted into Review Manager 5.3 for Meta-analysis. The included literature will be assessed for bias risk according to the bias risk assessment tools in the Cochrane Handbook for Systematic Reviews of Interventions. Funnel plots will be used to assess possible publication bias. Heterogeneity is tested by $I^2$ and $\chi^2$ tests, and the existence of heterogeneity is defined as $I^2 \geq 50\%$ and $P \leq 0.1$. If heterogeneity exists, the random-effect model will be used, and sensitivity analysis or subgroup analysis will be performed to further determine the source of heterogeneity.

**Funding:** This work was supported by School-level scientific research project of Southwest Medical University (Grant Number: 2021ZKQN042). The funder had and will not have a role in study design, data collection and analysis, decision to publish, or preparation of the manuscript.

**Competing interests:** The authors have declared that no competing interests exist.

**Abbreviations:** cGMP, cyclic guanosine monophosphate; NMDAR, N-methyl-D-aspartate receptor; NO, nitric oxide; RCTs, Randomized controlled trials.

## Prospero registration number

CRD42022341681.

## Introduction

Schizophrenia is a chronic persistent disease with high recurrence rate and high disability rate in the field of psychiatry [1], which is often accompanied by cognitive impairment and severe social function defects [2]. According to the World Health Organization (https://www.who.int/zh/news-room/fact-sheets/detail/mental-disorders), schizophrenia affects about 20 million people worldwide [3]. The average prevalence rates in Europe, America and Southeast Asia are 5.0 ‰, 4.2 ‰ and 3.7 ‰, respectively [4]. In China, epidemiological surveys in some provinces and cities show that the prevalence rate of schizophrenia has exceeded 5.0 ‰ [5]. In addition, the prevalence rate in areas with high economic development level is higher than that in areas with relatively backward economy [6]. More importantly, the burden of mental illness continues to increase in all countries around the world, which will further increase the social and economic family burden and have negative consequences for the health of individuals [7, 8].

The pathogenic mechanism of schizophrenia is not completely clear. Abnormal neurodevelopment mediated by genetic susceptibility, neurobiochemical factors and social environmental factors may lead to the occurrence of schizophrenia [9, 10]. Abnormal nitric oxide (NO) and glutamate signaling are involved in the onset of mental illness [11]. Patients with schizophrenia not only have N-methyl-D-aspartate receptor (NMDAR) dysfunction, but also change the expression levels of NO metabolites and cyclic guanosine monophosphate (cGMP) in body fluid [12, 13]. Therefore, targeted regulation of NMDA/NO/cGMP pathway may be one of the strategies for treating schizophrenia [14, 15].

Since schizophrenia is a group of syndromes that includes multiple symptoms and dysfunction, treatment of the disorder is difficult [16]. Although antipsychotic drugs can control patients' mental symptoms, they are relatively poor in the treatment of negative symptoms and cognitive impairment [2]. Thus, new drugs with high efficiency and safety need to be urgently developed [17]. Sodium nitroprusside is an emergency antihypertensive drug commonly used in clinic [18]. So far, sodium nitroprusside is being studied as a potential antipsychotic [19]. Sodium nitroprusside has been shown to be a NO donor and is considered a promising new compound for the treatment of schizophrenia [19]. Sodium nitroprusside could change DMDAR activity and increase cGMP expression level. It has been found that sodium nitroprusside can alleviate cognitive and memory deficits and social withdrawal induced by the ketamine (a NMDA receptor antagonist) in rats [20, 21]. More importantly, a clinical trial suggested that the symptoms of acute schizophrenia could be improved rapidly after intravenous injection of sodium nitroprusside [22].

Many clinical trials of sodium nitroprusside and schizophrenia have been published. However, the reports on its effectiveness are not uniform. We previously conducted a meta-analysis of the efficacy of intravenous sodium nitroprusside in schizophrenia and reached a negative conclusion [23]. New high-quality clinical trials of sodium nitroprusside in the treatment of schizophrenia have been published in recent years [24–26]. Since sodium nitroprusside does have the potential to treat schizophrenia, it is necessary to re-conduct the meta-analysis after the inclusion of these new clinical trials. This study will search the relevant clinical trials in this field, and conduct a systematic review and meta-analysis of the literature, so as to lay an evidence-based medicine foundation for the efficacy of sodium nitroprusside in the treatment of schizophrenia.

## Methods and analysis

### Study registration

The protocol has been registered with PROSPERO (https://www.crd.york.ac.uk/PROSPERO), registration number CRD42022341681, and reported according to Preferred Reporting Items for Systematic Review and Meta-Analysis Protocols (PRISMA-P) statement guidelines (S1 Checklist).

### Patient and public involvement

No patient or public participation in this study, so no ethical approval was required to conduct this study.

### Inclusion and exclusion criteria

**Inclusion criteria.** Studies that meet the following conditions will be included in the study: (1) patients diagnosed with schizophrenia by "*Diagnostic and Statistical Manual of Mental Disorders*" or "*International Classification of Diseases*" without restriction on their race, nationality, sex, and course of disease; (2) randomized controlled trials (RCTs) of sodium nitroprusside in the treatment of schizophrenia published in Chinese or English. (3) the intervention group was treated with sodium nitroprusside, while the control group was treated with normal saline or other treatment methods. Except not using sodium nitroprusside, the other intervention measures of the two groups were consistent as far as possible.

**Exclusion criteria.** The following studies will be excluded: (1) languages other than Chinese and English; (2) full text of the trial is not available; (3) the details in the table or figure are still not available by contacting the corresponding authors via e-mail; (4) lack of outcome measures; (5) abstracts, reviews, protocols, case reports or basic researches.

### Intervention/Control

The intervention group was treated with sodium nitroprusside, while the control group was treated with normal saline or other treatment methods. Except not using sodium nitroprusside, the other intervention measures of the two groups were consistent as far as possible.

### Outcome measures

Outcome measures are positive and negative Symptom Scale (PANSS), including PANSS positive Scale, PANSS Negative Scale, PANSS General psychopathology Scale, and 18-item Brief Psychiatric Rating Scale (BPRS-18). Besides, the description and evaluation of adverse reactions will be recorded.

### Search methods for identification of studies

RCTs of sodium nitroprusside in the treatment of schizophrenia will be searched through English databases (PubMed, Web of Science, Embase, and Cochrane Library) and Chinese databases (China Biology Medicine disc, VIP, WanFang Data, and CNKI). The method of "subject words + free words" will be used for searching, and the search words will mainly include: "schizophrenia," "sodium nitroprusside," and "nitroprusside." We will make reasonable adjustments according to different databases. The search strategy taking PubMed as an example is shown in Table 1.

**Table 1. Search strategy for PubMed.**

| Number | Search items |
|---|---|
| #1 | Schizophrenia [Mesh] OR Schizophrenia Spectrum and Other Psychotic Disorders [Mesh] OR Schizophrenia, Paranoid [Mesh] OR Schizophrenia, Disorganized [Mesh] OR Schizophrenia, Childhood [Mesh] OR Schizophrenia, Catatonic [Mesh] |
| #2 | schizophrenia OR schizo OR schizoid OR schizophrenic OR schizophreniac |
| #3 | #1 OR #2 |
| #4 | Nitroprusside [Mesh] OR sodium nitroprusside adenine complex [Supplementary Concept] |
| #5 | sodium nitroprusside OR nitroprusside |
| #6 | #4 OR # 5 |
| #7 | #3 AND #6 |

## Study selection and data extraction

Endnote X9 will be used to manage the searched articles, and the duplicate articles searched from each database will be deleted. Two researchers (XF and SW) will separately screen the literature and extract data in full accordance with the Cochrane Handbook for Systematic Reviews of Interventions. If there is disagreement, a third researcher will intervene (JL) and reach a conclusion after discussion. The flow chart will be drawn according to Preferred Reporting Items for Systematic Reviews and Meta-Analyses (PRISMA) (http://www.prisma-statement.org/), as shown in Fig 1.

The steps of literature screening and data extraction will be completed through the following three steps: (1) preliminary screening: browse the literature title and abstract, and exclude the irrelevant literature. (2) reading the full text: read the contents of the literature carefully and judge whether the literature meets the standard. (3) data extraction: the extraction

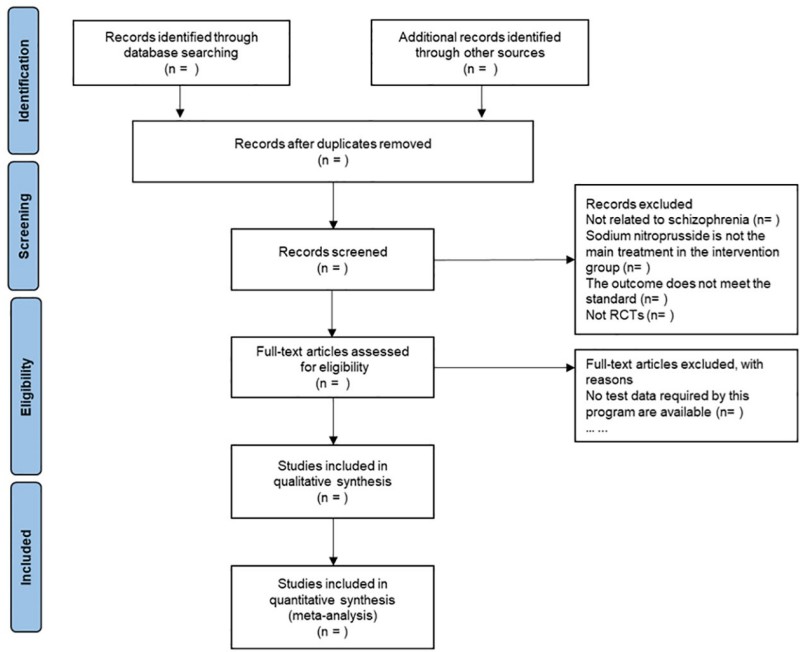

**Fig 1. PRISMA flow diagram showing the study selection process.**

contents are research title, authors, publication time, baseline and terminal characteristics, intervention methods, and follow-up time.

When there is no data in the literature, the data is obtained by contacting the first author or the correspondending author. If the data is still not available, the researchers will try using the GetData Graph Digitizer software (Australia) to extract the data. We will use the Excel spreadsheet to record the information of eligible articles, and all eligible studies will be included in the systematic review and meta-analysis.

## Assessment of risk of bias

The included literature will be assessed for bias risk according to the bias risk assessment tools in the Cochrane Handbook for Systematic Reviews of Interventions [27], including randomized methods, double-blind Settings, concealment protocols, selective reporting, data integrity, and other sources of bias. Two independent researchers (XF and SW) would conduct the assessment, and in case of disagreement, the third researcher (JL) would analyze the data extracted by both sides to find out the root cause of the data contradiction and draw a unified conclusion. In addition, funnel plots will be used to assess possible publication bias if at least 10 studies are included in the outcome indicator analysis.

## Data synthesis

The extracted data will be inputted into Review Manager 5.3 for Meta-analysis, and the test level is $\alpha_{(bilateral)} = 0.05$. The mean difference (MD) or standard mean difference (SMD) will be used as the effect size of the outcome measure depending on the synthesis of results. Heterogeneity is tested by $I^2$ and $\chi^2$ tests, and the existence of heterogeneity is defined as $I^2 \geq 50\%$ and $P \leq 0.1$. The random-effect model treats the regression coefficients of fixed effects models as random variables. When there is heterogeneity after the synthesis of various studies, the random-effect model can meet the requirements of the weighted average of multiple different population parameters of the combined effect value of meta-analysis [28, 29]. Moreover, sensitivity analysis or subgroup analysis will be performed to further determine the source of heterogeneity.

## Subgroup analysis and sensitivity analysis

On the premise of a sufficient number of studies, the subgroup analysis will be performed according to the type of schizophrenia, duration of disease, and age to explore the impact of these factors on prognosis. In addition, sensitivity analysis will be performed for the effects of studies of low quality, particularly heavy weights, or results that differed from other studies.

## Discussion

Schizophrenia is a serious mental illness that has a profound impact on individuals, families, and society [30]. Patients with schizophrenia have reduced cardiopulmonary function and muscle strength, and are at increased risk of cardiovascular disease, metabolic syndrome, obesity, hypertension, and hyperlipidemia [31]. Therefore, alleviating the increasing burden of schizophrenia and other mental diseases is an important research direction in the field of psychiatry.

Although the classical theory of dopamine neurotransmission is still being explored, recent evidence suggests that there are other abnormal factors in schizophrenia, including viral infection, cellular immune response, and dysregulation of glutamate nitric oxide cyclic guanosine phosphate network [32, 33]. The blockage of NMDAR in patients with schizophrenia can lead

to a decrease in NO production, resulting in a decrease in cGMP levels [33]. Sodium nitroprusside is a rapid, reliable, and easy-to-adjust antihypertensive drug, which can release NO and directly dilate arterioles and veins [34]. Thus, sodium nitroprusside has become a potential therapeutic drug in the treatment of schizophrenia.

In terms of molecular mechanism, a reasonable explanation is that sodium nitroprusside can regulate the NMDA/NO/cGMP pathway. Sodium nitroprusside can eliminate the behavioral effect of phencyclidine, an antagonist of glutamate NMDAR, which indirectly indicates that sodium nitroprusside may have the potential to treat mental disorders [35, 36]. However, the antipsychotic effect of SNP was initially observed and proposed in clinic [22]. In recent years, new high-quality trials on sodium nitroprusside in the treatment of schizophrenia have been published, which may lead to changes in the results of the previous meta-analysis.

Therefore, we believe it is necessary to conduct a systematic review and meta-analysis of the existing high-quality RCTs provide a basis for clinical decision-making. This protocol describes the specific steps and methods of the updated systematic review and meta-analysis. The findings will reveal the efficacy and safety of sodium nitroprusside in the treatment of schizophrenia. This study will provide new ideas for the drug treatment of schizophrenia. If positive results are obtained, it is very likely to optimize the existing drug treatment strategy. More importantly, medical staff can choose better treatment methods in clinical practice according to the findings of this study.

## Supporting information

**S1 Checklist. PRISMA-P (Preferred Reporting Items for Systematic review and Meta-Analysis Protocols) 2015 checklist: Recommended items to address in a systematic review protocol**\*.
(DOC)

## Author Contributions

**Conceptualization:** Shiqi Wang, Yaqian Gao, Yue Hu.

**Data curation:** Xinxing Fei, Shiqi Wang, Jiyang Li.

**Formal analysis:** Xinxing Fei, Shiqi Wang, Jiyang Li, Jianxiong Wang.

**Funding acquisition:** Jianxiong Wang, Yaqian Gao, Yue Hu.

**Methodology:** Xinxing Fei, Shiqi Wang, Yaqian Gao, Yue Hu.

**Software:** Xinxing Fei, Shiqi Wang.

**Writing – original draft:** Xinxing Fei, Shiqi Wang, Jiyang Li, Jianxiong Wang.

**Writing – review & editing:** Xinxing Fei, Yaqian Gao, Yue Hu.

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
