## [Decision Letter · Decision Letter 0]

11 Jan 2023

PONE-D-22-25733The efficacy and safety of sodium nitroprusside in the treatment of schizophrenia: protocol for an updated systematic review and meta-analysisPLOS ONE

Dear Dr. Hu,

Thank you for submitting your manuscript to PLOS ONE. After careful consideration, we feel that it has merit but does not fully meet PLOS ONE’s publication criteria as it currently stands. Therefore, we invite you to submit a revised version of the manuscript that addresses the points raised during the review process.

We look forward to receiving your revised manuscript.

Kind regards,

Vincenzo De Luca

Academic Editor

PLOS ONE

Journal Requirements:

Reviewers' comments:

Reviewer's Responses to Questions

**Comments to the Author**

1. Does the manuscript provide a valid rationale for the proposed study, with clearly identified and justified research questions?

Reviewer #1: Yes

2. Is the protocol technically sound and planned in a manner that will lead to a meaningful outcome and allow testing the stated hypotheses?

Reviewer #1: Yes

3. Is the methodology feasible and described in sufficient detail to allow the work to be replicable?

Reviewer #1: Yes

4. Have the authors described where all data underlying the findings will be made available when the study is complete?

Reviewer #1: No

5. Is the manuscript presented in an intelligible fashion and written in standard English?

Reviewer #1: No

6. Review Comments to the Author

You may also provide optional suggestions and comments to authors that they might find helpful in planning their study.

Reviewer #1: PONE-D-22-25733

Title: The efficacy and safety of sodium nitroprusside in the treatment of schizophrenia:

protocol for an updated systematic review and meta-analysis

Summary:

This is a systematic review protocol of a planned review of studies on sodium nitroprusside in the treatment of schizophrenia. The review proposes that it will use updated literature that includes studies that have not been included in reviews to date. The authors outline the analytic approach to assessing publication bias, heterogeneity and sensitivity analyses.

Major Comments:

Overall the contribution to the field and need for the review is clear. Suggestions are made throughout for the need for academic language to be used, and to add references throughout.

Furthermore, there appear to be issues with tenses. If this is a protocol my understanding is that it should be written framing around what will be done, however the authors are writing in past tense. Has this review then already been completed prior to publishing the protocol? Please be clear about the stage of the work, and be sure the tenses used in the writing are clear and consistent throughout the paper.

Minor comments:

Abstract.

Spell out NO donor on first use, the NO acronym

The authors note that qual and quant analysis is needed. Do they plan to do qual analysis? If not they should not mention it here

Background:

1. The statement below needs a reference, and please don’t use the word “we”

Therefore, we may be able to treat schizophrenia

by regulating the NMDA-NO-cGMP pathway.

2. Please change urgent to urgently and include a reference

Thus, the development of efficient and safe new drugs has become an

urgent need.

3. For the below, please remove “As we all know,”

As we all know, sodium nitroprusside is an emergency antihypertensive drug

commonly used in clinic

4. For the below please add reference

Up to now, sodium nitroprusside is being studied as a potential

Antipsychotic

5. For the below please change “our” to “this”

Our study will search the relevant clinical trials in this field

Methods

1. For the below should say “patients diagnosed with schizophrenia”

patients

diagnosed as schizophrenia by

2. For the below, the sentence is not clear, it is in past tense, maybe needs to be future tense?

Except not using sodium nitroprusside, the other intervention

measures of the two groups were consistent as far as possible.

3. Exclusion criteria—there is a note about contacting the author—what is the procedure for this contact? How will the authors do this?

4. For the below, again is this already completed? If not, then if this is a protocol it should be future tense about what will be done?

Besides, description and evaluation of adverse

reactions were recorded

5. Again, issues of tense here, mixing past and future

The method of

"subject words + free words" will be used for searching, and the search words mainly

included: "schizophrenia," "sodium nitroprusside," and "nitroprusside."

6. For the below what does mainly completed mean? Do you mean solely/completely/only completed through this?

“The steps of literature screening and data extraction are mainly completed through the

following three steps:”

7. For the below should say “ we will try using the”

If the data is still not available, try using the GetData Graph

Digitizer software (Australia) to extract the data.

8. Under this heading : Assessment of risk of bias

Please add a reference to the Cochrane Handbook

9. Can you say. More about this? How will the 3rd person be able to help reach agreement?

Two independent researchers (XF and SW)

would conduct the assessment, and in case of disagreement, a third researcher (JL) would

intervene to reach a unified conclusion after participating in the discussion.

10. For the below, explain the random effect model, how it works, what its purpose is and why it is appropriate and add a reference

random-effect model will be used

11. subgroup analysis:

If necessary,

*how will you determine “if necessary” ? is it based on the number and diversity of populations in the studies identified? Please explain

Discussion

1. It does not seem to me that the discussion section expands much more beyond repeating the background. Can the authors add some information regarding how they expect the review to inform potential next steps in clinical practice and policy?

7. PLOS authors have the option to publish the peer review history of their article (what does this mean?). If published, this will include your full peer review and any attached files.

Reviewer #1: No

---

## [Author Response · Author response to Decision Letter 0]

12 Feb 2023

Editorial Board

PLOS ONE

March 28, 2022

Dear Editors and Reviewers, 

We are resubmitting our revised manuscript entitled “The efficacy and safety of sodium nitroprusside in the treatment of schizophrenia: protocol for an updated systematic review and meta-analysis” by Fei et al. to PLOS ONE. We have addressed your helpful comments and suggestions as outlined below and revised the manuscript accordingly. We hope that the manuscript is ready to be published in PLOS ONE.

Response to Reviewer 1’s comments:

Reviewer #1: PONE-D-22-25733

Title: 

The efficacy and safety of sodium nitroprusside in the treatment of schizophrenia:

protocol for an updated systematic review and meta-analysis

Summary:

This is a systematic review protocol of a planned review of studies on sodium nitroprusside in the treatment of schizophrenia. The review proposes that it will use updated literature that includes studies that have not been included in reviews to date. The authors outline the analytic approach to assessing publication bias, heterogeneity and sensitivity analyses.

Major Comments:

Overall the contribution to the field and need for the review is clear. Suggestions are made throughout for the need for academic language to be used, and to add references throughout. Furthermore, there appear to be issues with tenses. If this is a protocol my understanding is that it should be written framing around what will be done, however the authors are writing in past tense. Has this review then already been completed prior to publishing the protocol? Please be clear about the stage of the work, and be sure the tenses used in the writing are clear and consistent throughout the paper.

Response: 

Thank you for your comments. We have revised the language of the manuscript, regulated the use of academic language, and added references based on your suggestions. In addition, we asked a native speaker to edit the manuscript for language. This protocol was completed before the systematic review, so we agree with you that it is mainly written in the future tense in the section of method.

Minor comments:

Abstract

Spell out NO donor on first use, the NO acronym

Response: Thanks for your suggestion, we have added the full name of NO.

- Sodium nitroprusside is a nitric oxide (NO) donor… (Page 4)

The authors note that qual and quant analysis is needed. Do they plan to do qual analysis? If not they should not mention it here.

Response: 

Thanks for your suggestion, we deleted it. And we have rephrased this sentence.

- It is necessary to re-conduct the meta-analysis after the inclusion of these new clinical trials. (Page 4)

Background:

1. The statement below needs a reference, and please don’t use the word “we”

Therefore, we may be able to treat schizophrenia by regulating the NMDA-NO-cGMP pathway.

Response: 

Thanks to your suggestion, we have deleted the "we" and added references.

- Therefore, targeted regulation of NMDA/NO/cGMP pathway may be one of the strategies for treating schizophrenia. (Page 5)

-Reference 14 and 15 (Page 14)

2. Please change urgent to urgently and include a reference

Thus, the development of efficient and safe new drugs has become an urgent need.

Response: 

We have rephrased this sentence and added a reference.

- Thus, new drugs with high efficiency and safety need to be urgently developed. (Page 5)

- Reference 17 (Page 14)

3. For the below, please remove “As we all know,” 

As we all know, sodium nitroprusside is an emergency antihypertensive drug commonly used in clinic

Response: 

We are sorry to have affected your reading experience. We revised it immediately.

- Sodium nitroprusside is an emergency antihypertensive drug commonly used in clinic. (Page 5)

4. For the below please add reference 

Up to now, sodium nitroprusside is being studied as a potential Antipsychotic

Response: 

According to your suggestion, we have added the reference.

- reference 19 (Page 14)

5. For the below please change “our” to “this”

Our study will search the relevant clinical trials in this field

Response: 

Thanks for your advice, we have revised it. We have checked the manuscript and tried to avoid writing in the first person.

- This study will search the relevant clinical trials in this field, and conduct a systematic review and meta-analysis of the literature, … (Page 6)

Methods

1. For the below should say “patients diagnosed with schizophrenia”

patients diagnosed as schizophrenia by

Response: 

Thank you for your suggestion. We have revised it. 

- patients diagnosed with schizophrenia by "Diagnostic and Statistical Manual of Mental Disorders" or … (Page 7)

2. For the below, the sentence is not clear, it is in past tense, maybe needs to be future tense?

Except not using sodium nitroprusside, the other intervention measures of the two groups were consistent as far as possible.

Response: 

We apologize for interrupting your reading experience. We think it should be past tense here, because we will include randomized controlled trials that have been carried out. These randomized controlled trials have been conducted and published.

3. Exclusion criteria—there is a note about contacting the author—what is the procedure for this contact? How will the authors do this?

Response: 

We are sorry we did not explain it clearly. In fact, when the data cannot be obtained through the article, we will contact the corresponding authors involved in the study by e-mail as much as possible. We have also redescribed this part.

- the details in the table or figure are still not available by contacting the corresponding authors via e-mail. (Page 7)

4. For the below, again is this already completed? If not, then if this is a protocol it should be future tense about what will be done?

Besides, description and evaluation of adverse reactions were recorded

5. Again, issues of tense here, mixing past and future

The method of "subject words + free words" will be used for searching, and the search words mainly included: "schizophrenia," "sodium nitroprusside," and "nitroprusside."

Response to No.4 and No.5: 

We apologize for interrupting your reading experience, and we have corrected the tense immediately.

- Besides, the description and evaluation of adverse reactions will be recorded. (Page 7)

- The method of "subject words + free words" will be used for searching, and the search words will mainly include: "schizophrenia," "sodium nitroprusside," and "nitroprusside." (Page 8)

6. For the below what does mainly completed mean? Do you mean solely/completely/only completed through this?

“The steps of literature screening and data extraction are mainly completed through the following three steps:”

Response: 

We are sorry that we did not explain it clearly. In fact, data extraction will be carried out in full accordance with the Cochrane Handbook for Systematic Reviews of Interventions. Therefore, our steps will follow exactly the process we described. We have deleted "mainly" to avoid ambiguity.

- The steps of literature screening and data extraction will be completed through the following three steps: … (Page 8-9)

7. For the below should say “we will try using the”

If the data is still not available, try using the GetData Graph Digitizer software (Australia) to extract the data.

Response: 

Thanks for your suggestion, we have revised the tenses immediately.

- If the data is still not available, the researchers will try using the GetData Graph Digitizer software (Australia) to extract the data. (Page 9)

8. Under this heading: Assessment of risk of bias

Please add a reference to the Cochrane Handbook

Response: 

According to your suggestion, we have added the reference.

- reference 27 (Page 14)

9. Can you say. More about this? How will the 3rd person be able to help reach agreement?

Two independent researchers (XF and SW) would conduct the assessment, and in case of disagreement, a third researcher (JL) would intervene to reach a unified conclusion after participating in the discussion.

Response: 

We are sorry we did not explain it clearly. At present, most meta-analysis guidelines suggest that the degree of bias of data extraction by two individuals is lower than that of two extracts by one person. When there is disagreement, a decision needs to be discussed or decided by a third person. In our study, the third person will analyze the extracted data of two independent researchers on the basis of using the bias risk assessment tools to find out the root cause of data contradictions.

- Two independent researchers (XF and SW) would conduct the assessment, and in case of disagreement, the third researcher (JL) would analyze the data extracted by both sides to find out the root cause of the data contradiction and draw a unified conclusion. (Page 9)

10. For the below, explain the random effect model, how it works, what its purpose is and why it is appropriate and add a reference

random-effect model will be used

Response: 

Thank you for your advice. We explain the random-effect model and add references.

- The random-effect model treats the regression coefficients of fixed effects models as random variables. When there is heterogeneity after the synthesis of various studies, the random-effect model can meet the requirements of the weighted average of multiple different population parameters of the combined effect value of meta-analysis. (Page 9)

11. subgroup analysis:

If necessary,

*how will you determine “if necessary” ? is it based on the number and diversity of populations in the studies identified? Please explain

Response: 

We are sorry we did not explain it clearly. At this time, we cannot determine the number of studies that will be included and the characteristics of the participants (such as sex, age, race, etc.). If the final number of included studies is too small to conduct subgroup analysis, we have to give up. In order to avoid unnecessary misunderstanding, we have redescribed this sentence. 

- On the premise of a sufficient number of studies, the subgroup analysis will … (Page 10)

Discussion

1. It does not seem to me that the discussion section expands much more beyond repeating the background. Can the authors add some information regarding how they expect the review to inform potential next steps in clinical practice and policy?

Response: 

Thank you for your suggestion, and it enrich the discussion section of our manuscript.

- This protocol describes the specific steps and methods of the updated systematic review and meta-analysis. The findings will reveal the efficacy and safety of sodium nitroprusside in the treatment of schizophrenia. This study will provide new ideas for the drug treatment of schizophrenia. If positive results are obtained, it is very likely to optimize the existing drug treatment strategy. More importantly, medical staff can choose better treatment methods in clinical practice according to the findings of this study. (Page 11)

7. PLOS authors have the option to publish the peer review history of their article (what does this mean?). If published, this will include your full peer review and any attached files.

Do you want your identity to be public for this peer review? For information about this choice, including consent withdrawal, please see our Privacy Policy.

Reviewer #1: No

Thank you very much for reviewing our manuscript.

Yours Sincerely,

Yue Hu

---

## [Decision Letter · Decision Letter 1]

6 Mar 2023

The efficacy and safety of sodium nitroprusside in the treatment of schizophrenia: protocol for an updated systematic review and meta-analysis

PONE-D-22-25733R1

Dear Dr. Hu,

We’re pleased to inform you that your manuscript has been judged scientifically suitable for publication and will be formally accepted for publication once it meets all outstanding technical requirements.

Kind regards,

Vincenzo De Luca

Academic Editor

PLOS ONE

Additional Editor Comments (optional):

Reviewers' comments:

Reviewer's Responses to Questions

**Comments to the Author**

1. Does the manuscript provide a valid rationale for the proposed study, with clearly identified and justified research questions?

Reviewer #1: Yes

2. Is the protocol technically sound and planned in a manner that will lead to a meaningful outcome and allow testing the stated hypotheses?

Reviewer #1: Yes

3. Is the methodology feasible and described in sufficient detail to allow the work to be replicable?

Reviewer #1: Yes

4. Have the authors described where all data underlying the findings will be made available when the study is complete?

Reviewer #1: Yes

5. Is the manuscript presented in an intelligible fashion and written in standard English?

Reviewer #1: Yes

6. Review Comments to the Author

You may also provide optional suggestions and comments to authors that they might find helpful in planning their study.

Reviewer #1: No additional comments at this time, the authors have addressed all of my comments. Thank you of rthe opportunity to review this manuscript.

7. PLOS authors have the option to publish the peer review history of their article (what does this mean?). If published, this will include your full peer review and any attached files.

Reviewer #1: **Yes: **Heather Palis

---

## [Editor Report · Acceptance letter]

10 Mar 2023

PONE-D-22-25733R1 

The efficacy and safety of sodium nitroprusside in the treatment of schizophrenia: protocol for an updated systematic review and meta-analysis 

Dear Dr. Hu:

I'm pleased to inform you that your manuscript has been deemed suitable for publication in PLOS ONE. Congratulations! Your manuscript is now with our production department. 

Kind regards, 

on behalf of

Dr. Vincenzo De Luca 

Academic Editor

PLOS ONE